# The European Map Butterfly *Araschnia levana* as a Model to Study the Molecular Basis and Evolutionary Ecology of Seasonal Polyphenism

**DOI:** 10.3390/insects12040325

**Published:** 2021-04-06

**Authors:** Arne Baudach, Andreas Vilcinskas

**Affiliations:** 1Institute for Insect Biotechnology, Justus-Liebig University of Giessen, 35392 Giessen, Germany; arnebaudach@gmail.com; 2Department of Bioresources, Fraunhofer Institute for Molecular Biology and Applied Ecology, Ohlebergsweg 12, 35392 Giessen, Germany

**Keywords:** evolutionary ecology, hormones, epigenetics, polyphenism, phenotypic plasticity, microRNAs, metamorphosis, diapause, *Araschnia levana*

## Abstract

**Simple Summary:**

The European map butterfly looks different in spring and summer due to day length and temperature. If the butterfly’s caterpillars receive 16 h of light per day, the resulting butterfly hatches a few weeks later with blackish wings. However, if caterpillars receive less than 15.5 h of daylight, many overwinter as pupae. In the following spring, butterflies have orange wings. Overwintering and wing color are decided by hormones. If a certain hormone is released in the first days after the caterpillar has become a pupa, no overwintering takes place, and the wings are black. If this hormone is released later, overwintering occurs, and the wings are orange. Different genes are activated to make either of those two options happen. They guide what happens during overwintering and how long it lasts but also how the butterfly looks once it hatches. We do not yet fully understand how the caterpillars count the amount of light they receive and how this information leads to the differences described above. In addition, the butterfly’s whole body and its immune system are different in the two color types. Here we discuss how the butterfly probably makes these changes happen and which role the environment plays.

**Abstract:**

The European map butterfly *Araschnia levana* is a well-known example of seasonal polyphenism. Spring and summer imagoes exhibit distinct morphological phenotypes. Key environmental factors responsible for the expression of different morphs are day length and temperature. Larval exposure to light for more than 16 h per day entails direct development and results in the adult f. *prorsa* summer phenotype. Less than 15.5 h per day increasingly promotes diapause and the adult f. *levana* spring phenotype. The phenotype depends on the timing of the release of 20-hydroxyecdysone in pupae. Release within the first days after pupation potentially inhibits the default “*levana*-gene-expression-profile” because pre-pupae destined for diapause or subitaneous development have unique transcriptomic programs. Moreover, multiple microRNAs and their targets are differentially regulated during the larval and pupal stages, and candidates for diapause maintenance, duration, and phenotype determination have been identified. However, the complete pathway from photoreception to timekeeping and diapause or subitaneous development remains unclear. Beside the wing polyphenism, the hormonal and epigenetic modifications of the two phenotypes also include differences in biomechanical design and immunocompetence. Here, we discuss research on the physiological and molecular basis of polyphenism in *A. levana*, including hormonal control, epigenetic regulation, and the effect of ecological parameters on developmental fate.

## 1. Introduction

The phenotype of an organism is dependent on the genome and its epigenetic regulation, based on a combination of cellular memory and interactions with the environment [1]. The term phenotypic plasticity thus refers to the ability of an organism to generate different phenotypes from the same genotype under different environmental conditions [2]. Phenotypic plasticity often facilitates adaptive changes by increasing phenotypic diversity in response to environmental challenges. Polyphenism is a special case of phenotypic plasticity in which the outputs are discrete and discontinuous, resulting in multiple distinct phenotypes from the same genetic background [3]. Some of the most striking examples of polyphenism in animals include sex determination in reptiles and fish regulated by temperature and social factors [4,5], the defense polyphenism in cladocerans [6], the sexual and wing polyphenism in aphids [7], and seasonal polyphenism in butterflies [8].

In 1758, Carl Linnaeus described two apparently distinct butterfly species, which he named *Papilio levana* and *P. prorsa*, but subsequent field observations and breeding experiments revealed them to be seasonal variants of the same bivoltine species: The European map butterfly *Araschnia levana* [9,10]. The dorsal wing of the spring generation (*A. levana* f. *levana*) is orange to reddish-brown (basic coloration) with black spots, some white dots, and a bluish dotted rim on the posterior of the hindwing (Figure 1). Conversely, the dorsal wing of the summer generation (*A. levana* f. *prorsa*) is brownish to bluish-black with a prominent white band (featuring varying degrees of melanization) located basally with respect to 1–3 apical orange bands (Figure 1). Many variations between these two phenotypes have been reported, and they are best described as a spectrum. Some of these occur naturally as *A. levana* f. *porima*, with patterning and coloration appearing intermediate between *levana* and *prorsa.* However, most are the result of experimental manipulation [10,11]. The elements of the wing underside are consistent between morphs and gave rise to the genus name *Araschnia*. The basic coloration is a darkish brown with another prominent whitish band separating the apical and basal parts of the wings. The veins have whitish scales and, with their fine-crossed connections, form a grid that is reminiscent of a spider web or map.

The remarkable change from vernal to estival phenotypes has attracted much interest from scientists. Naturalists initially tested the effect of temperature to determine the biophysical basis of the phenomenon [12,13,14], but their findings were also contested by contemporaries [15]. Alternatively, strict cyclic heritability was proposed to explain the rhythmic alternation of spring and summer phenotypes [16]. The first studies on seasonal changes in insects were conducted by Marcovitch in the 1920s, who revealed a connection between the appearance of sexual forms in relation to day length in aphids [17]. However, it was not until the 1950s that the true ecological parameters responsible for the polyphenic shift in *A. levana* were revealed. Müller (1955) demonstrated that larvae of either (parental) generation developed directly to become subitaneous pupae and then displayed the adult *prorsa* (long-day or summer) phenotype if they were exposed to light for more than 16 h per day, whereas all larvae became diapause pupae and thus displayed the adult *levana* (short-day or spring) phenotype if they were exposed to light for less than 8 h per day [18]. Following this breakthrough discovery, the physiological basis of polyphenism was the next problem to be addressed [19]. Another species, the Satyrid *Bicyclus anynana*, seasonally exhibits striking differences in the ventral wing pattern too [20]. In this model species, the plasticity of the eyespot size is mostly regulated by temperature, which—in the wandering stage—leads to changing titers of the hormone 20-hydroxyecdysone (20E) [21]. If the 20E signaling is manipulated at that specific time in development, eyespot size can easily be modified. However, for *A. levana*, it remained unclear for over 60 years how the physiological switches were likely regulated at the molecular level to orchestrate the manifestation of two discrete phenotypes [22,23]. In this review, we discuss research conducted to address these issues, including the role of different ecological parameters and their effect on phenotype, where known. We consider the hormones that mediate the switch from *levana* to *prorsa* morphs, and the epigenetic framework in which they operate. Finally, we briefly discuss phenotypic distinctions other than wing coloration (such as differential immunocompetence), draw some conclusions based on our current knowledge, and identify potential directions for future research.

## 2. Environment and Phenotype

### 2.1. Photoperiodism and Temperature

The longstanding assumption that temperature was the abiotic factor responsible for the phenotypic switch in *A. levana* was elegantly refuted by Müller (1955). Before presenting his findings, he remarked on the matter of temperature: “*In the field, it cannot be the cause, as it is on average about the same during the crucial developmental phases of the two generations*” [18]. In his experiment, he assigned offspring from both generations into four treatment groups. He then reared one group from each parental generation under two separate light regimes but otherwise identical conditions, in particular, at the same temperature. The two groups exposed to more than 16 h of light per day developed exclusively into subitaneous pupae and thus into the *prorsa* form, whereas the two groups reared under short-day conditions (8 h of light per day) invariably developed into diapause pupae and thus into the *levana* form, regardless of the parental generation [18].

Because the potentially modifying influence of temperature was still unclear at this point, Müller subsequently investigated the effects of the distinct light regimes at two different temperatures: 20 and 30 °C [24]. At 20 °C, all larvae developed into the *levana* form if exposed to fixed light regimes of 4–15 h per day, whereas there was an inverse relationship between light duration and the proportion of *prorsa* individuals in the same photoperiodic range when larvae were reared at 30 °C. Specifically, longer photoperiods led to a steady decline in the proportion of *prorsa* individuals. When the day length was 6 h, the proportions of *prorsa* and *levana* adults were approximately equal, but when the day length was 12 h the ratio was 3% *prorsa* to 97% *levana*. However, at both temperatures, a switch occurred between day lengths of 15.5 and 16.5 h. More than 16.5 h resulted in the complete inhibition of *levana* development, yielding 100% *prorsa* adults. When the photoperiod falls below 15–16 h (daylight lasts for 15.5 h between the middle of May and late July, in this study at app. 51°8′ N 11°1′ E), the temperature is, therefore, used as an additional cue to determine whether direct development or diapause is preferred.

Later work showed that the critical photoperiod is longer at lower temperatures, with a temperature regime of 15 °C shifting the photoperiod needed for direct development towards longer days [25]. Exposure to 16 h of daylight at this temperature still committed little more than half of all larvae to direct development. These findings indicate that longer photoperiods are required to induce direct development at lower temperatures, whereas shorter photoperiods are sufficient at higher temperatures, although the latter only applies to photoperiods of less than 12 h. In an ecological context, this means that warmer temperatures tip the risk–benefit ratio in favor of direct development (betting on continued beneficial conditions), whereas cooler temperatures have the opposite effect (betting on an overwintering strategy). The increase in critical day length is more likely to prevent direct development and consequently the formation of a potential third butterfly generation. In Central Europe, this reflects conditions in the wild, where *prorsa* larvae develop from mid-May to mid-July at day lengths of at least 16.5 h and mean temperatures of 15–18 °C [18,24,25] (Figure 1). In contrast, *levana* larvae develop in August and September, when the mean temperature is initially ~19 °C but quickly declines to ~11 °C by the beginning of October [18]. Day length during the same period declines from 15.5 to 11.5 h (Figure 1). This suggests that the photoperiod takes precedence as the key climate predictor with the ultimate decision-making role, but it can be modified and fine-tuned by prevailing temperatures.

The development of subitaneous or diapause pupae depends on the day length in the mid (but not early or late) larval stages, with a critical photoperiod of ~15.5 h [24,26]. These findings have been modified by a more recent study [27], although direct comparisons are not possible because the data provided in the original studies are not precise. Larvae were reared at 23 °C under short-day conditions (12 h photoperiod) or long-day conditions (20 h photoperiod), and subsets were transferred between conditions in both directions in each of the five instars. Transfer from long-day to short-day conditions during the first three larval stages invariably led to diapause development, whereas the transferred fourth-instar larvae yielded ~40% *prorsa* adults and transferred final-instar larvae yielded 100% *prorsa* adults [27]. A fixed number of long days (18 h photoperiod) is necessary for direct development, representing up to half of the entire larval development period (~23 days) at 20 °C [25]. This indicates that there is a point of commitment during the fourth-instar stage beyond which diapause development is no longer possible, an advantageous strategy given the additional preparations needed to survive winter, such as general physiological changes, the formation of denser tissues, and the thickening of the cuticle [27]. On the other hand, it also implies that natural selection favors a decision made late in larval development, when larvae have the most current information about their position in the season. Friberg and colleagues also showed that transfer from short-day to long-day conditions during the first four larval stages led reproducibly to 100% *prorsa* adults, and even when switched during the final larval stage, there was still a 50% likelihood of *prorsa* development. In nature, longer day lengths correspond to spring and early summer (Figure 1). In years with early-season high temperatures, imagoes may emerge ahead of time, as reported in 1990 in the south-west of Germany [28]. After mating and oviposition, larval development may, therefore, start when day lengths are below the critical photoperiod for *prorsa* development of >15.5 h. This threshold is likely to be even higher, given that the mean early-season temperatures are still comparatively low even in unusually warm years and low temperatures require longer day lengths in order to achieve direct development (contrast with the modification of critical day length by low temperatures, as discussed above). In such cases, it would, therefore, be beneficial if larvae were able to identify and respond to a “switch from short days to long days” and accordingly favor the subitaneous pathway over diapause throughout larval development.

Temperature can also modify photoperiod effects at later stages of development. As with many nymphalids and other butterflies, the influence of higher or lower temperatures during early pupal development can lead to a brightening or darkening of wing color patterns in both *Araschnia* generations. However, a complete change from *levana* to *prorsa* or vice versa is not possible [29]. For subsequent development, only the temperature is relevant because both pupae and imagoes are profoundly insensitive to day length. During diapause, pupae must undergo a cool period (0–10 °C) lasting at least 3 months before eclosion can be induced by spring temperatures of 12–24 °C [26].

### 2.2. Food Quality

Not only day length and temperature change seasonally but also availability, composition, and quality of food sources. These may, therefore, have an impact on the phenotype in their own right. In both generations, *A. levana* larvae are strictly monophagous and feed exclusively on leaves of the stinging nettle *Urtica dioica*. However, the nutritional quality of this plant deteriorates as the nettle matures and thus also varies seasonally [30,31,32]. Seasonal variations in adult food sources are also conceivable, suggesting that food quality may influence the fitness of *A. levana*, which may contribute to polyphenism. Access to carbohydrates, nitrogen, and amino acids across juvenile and adult stages is known to influence the adult size, longevity, and fecundity in many lepidopteran species [30,33]. Adult female *A. levana* f. *prorsa* reared on a low-quality larval diet preferred a high-quality nectar mimic containing both essential and non-essential amino acids [33]. The authors proposed that such a preference implies that adult resources are more important when larval reserves are poor and that butterflies may compensate for adverse larval conditions by selective adult feeding. They further demonstrated a negative relationship between emergence mass and amino acid preference regardless of the larval diet, with amino acid preference diminishing as female mass increased. Interestingly, male larvae did not display any food preference regardless of the larval feeding state or emergence mass, suggesting this trait is linked to female fecundity or sex determination.

A follow-up study tested whether the use of nectar amino acids by adult female *A. levana* f. *prorsa* increased fecundity [30]. The authors evaluated the effect of low-quality and high-quality larval diets (based on the *U. dioica* leaf nitrogen content) combined with adult high-quality or low-quality nectar mimics (with or without amino acids). They compared the effects of these four diets on multiple fecundity parameters, including the number of eggs laid, egg mass, longevity, and hatching rate. Female emergence mass (reflecting larval food quality), adult nectar diet, and the amount of nectar consumed all had significant effects on the number of eggs laid. Individuals produced fewer eggs only if they were reared on a low-quality larval diet and, as adults, nectar lacking amino acids. The egg number was on par in the three other groups, including individuals reared on a low-quality larval diet but switched to the amino acid-rich high-quality nectar as adults. However, the egg mass, carbon to nitrogen (C/N) ratio, and hatching rate stayed the same even under adverse conditions, indicating that the fitness cost is purely quantitative. Furthermore, there was no significant effect on longevity in any treatment group, and the C/N ratio of abdomens from female specimens did not vary. The authors reported a positive correlation between butterfly emergence mass and the total number of eggs laid. Emergence mass depends on larval nutrition, and poor-quality foliage is typical for wild *A. levana* larvae feeding later in the year. This supports the presence of polyphenism in addition to compensatory adult feeding.

Differences have been reported in the body composition of pupae and imagoes depending on sex and/or light regime applied during the rearing of larvae [32]. Groups were reared with a 16 h photoperiod representing the spring and early summer conditions of the subitaneous *prorsa* generation and a 12 h photoperiod representing the late summer and early autumn conditions of the diapausing *levana* generation [32]. The water content of male and female pupae was lower in *levana* than *prorsa* individuals, and the male *levana* pupae also featured lower concentrations of lipids, but there were no significant differences in sugar or protein content and dry weight. Interestingly, the differences were not apparent in the adults. Indeed, the water content was higher in *levana* than *prorsa* imagoes at eclosion, and there was no seasonal difference in lipid content. These findings were put forward as evidence for phenotypic traits associated with pupal diapause and overwintering [32]. For example, diapausing *levana* pupae may be adapted specifically to prevent further dehydration, allowing them to enter pupation with a lower water content, protecting them against freezing. The authors did not report an increase in sugar content, but suggested that more subtle changes in sugars related to freeze tolerance in *levana* pupae may have been overlooked. This is because their methods were insensitive to variations in sugars associated with regular metabolism (such as glucose) and sugars that act as cryoprotectants (such as trehalose) but also polyols such as glycerol with a similar role.

The *levana* adults weighed less than their *prorsa* counterparts, which may reflect the metabolic cost of diapause and overwintering [32]. Notably, the lower body mass primarily affected the adult head, thorax, and wings. Because *levana* adults also emerged with lower protein concentrations, the flight musculature (and, by extension, flight capacity) is likely to be affected, consistent with findings in field-caught butterflies [32,34,35]. Both morphs in the study were fed on the same larval diet of freshly-picked *U. dioica* collected between July and August, representing the mediocre to poor food quality typically encountered by *levana* larvae. The study design is, therefore, likely to have masked any polyphenic effects related to differences in food quality.

Recently, a study by Esperk and Tammaru (2021) reported comparisons of various parameters of larval growth schedules in a 2 × 2 × 2 crossed design with photoperiod, temperature, and host plant quality as the varied factors [36]. Specifically (among other findings), they showed that *levana* larvae spent more time in both final and penultimate larval instars. Lower (but not higher) temperatures, also promoted lower *levana* pupal masses. In contrast, *prorsa* larvae displayed higher growth rates—a pattern that was consistent across different rearing conditions, sexes, and larval instars. These authors concluded that their findings demonstrated that the between-generation differences in development have a significant element of anticipatory plasticity and thus should be considered adaptive.

The latter study elegantly demonstrated how the approaches of the other studies described above could be combined to disentangle further the responses to food regimes as well as polyphenic adaptations to predictable seasonal variations in nutrition.

## 3. Hormonal and Epigenetic Control of the Phenotype

### 3.1. Hormones

Having determined the role of photoperiod, temperature, and (to a certain extent) other environmental factors on polyphenism in *A. levana*, researchers turned their attention to the translation of these signals at the physiological level. In other lepidopterans, adult development in both subitaneous and diapausing pupae is known to be triggered by the release of the prothoracicotropic hormone (PTTH) from the brain, followed by the release of ecdysone from the prothoracic glands [37]. However, preliminary experiments in *A. levana* and the closely related species *A. burejana* indicated that morph determination depended exclusively on the timing of ecdysteroid release [38,39].

In *A. levana*, subitaneous development is characterized by the accumulation of ecdysteroids in the fourth-instar and final-instar larvae, leading to an earlier *prorsa* pupal molt [40]. In subitaneous *prorsa* pupae, ecdysteroid levels peak at the mid-stage and decrease towards the imaginal molt, whereas the ecdysteroid content of diapausing *levana* pupae is low at the time when *prorsa* pupae develop into adults. In both morphs, the titer of juvenile hormone (JH) is high in the middle of the fourth-instar stage but declines thereafter, falling to undetectable levels two days into the final larval stage. In *levana*, JH remains undetectable for the rest of larval development, but in *prorsa* larvae there are two JH peaks before the pupal molt [40]. It seems plausible that these high JH titers trigger ecdysteroid release from the prothoracic gland during the pupal stage in subitaneous *prorsa* individuals (Figure 2).

Koch (1987) showed that when adult development was initiated by 20-hydroxyecdysone injection 3 days after pupation, the *prorsa* phenotype was produced. Conversely, when adult development was induced after 5 days or even post diapause, the *levana* form emerged [42]. In consecutive studies, adult *prorsa* development was shown to begin as early as 1 day after pupation, followed by a transitional period of another 2 or 3 days during which, if triggered, intermediate wing coloration would develop, followed thereafter solely by *levana* development. In the latter case, the ecdysteroid level remains low for a cold period lasting more than 3 months, then rises to induce the development of *levana* adults [19,26,40] (Figure 2). These findings suggest that the mination of wing color pattern changes gradually during the first week of pupal life, beginning with the summer morph and then changing into the spring morph [42].

In other words, the ultimate phenotype depends on the time at which adult development is initiated by the release of 20-hydroxyecdysone. The authors also found that both phenotypes could form even in pupae from which the brain-corpora cardiaca-allata complex had been surgically removed, as long as ecdysteroids were injected at the appropriate time. Accordingly, no brain-derived factors such as PTTH are required, and the polyphenism in *A. levana* is exclusively regulated by the timing of the 20-hydroxyecdysone release. Koch and Bückmann (1987) also showed that both seasonal wing phenotypes, as well as intermediary forms, can (at least experimentally) be produced by pupae that have experienced either long or short larval photoperiods. They concluded that seasonal wing coloration is not immediately affected by the action of day length and that photoperiod only governs pupal diapause or subitaneous development. They postulated a common regulatory mechanism based on the timing of ecdysteroid secretion, which thereby specifies the duration of the pupal stage as well as the adult wing phenotype.

### 3.2. Circadian Clocks and Epigenetics

In physiological terms, *A. levana* larvae must quantify the light they receive in some manner in order to realize one of the two seasonal phenotypes. It is unclear precisely how this photoperiodic information is perceived, stored, and acted upon during the development of insects [43], although a theoretical framework has been proposed [44]. The first step is light perception by photoreceptors, which probably involves stemmata or extraretinal photoreception, followed by signal transduction [45]. In the brain, a photoperiodic clock responsible for timekeeping measures the hours of darkness in the diurnal cycle, and this mechanism appears to be directly or indirectly sensitive to the temperature. Third, a counter keeps track of the number of times the “long-day threshold” has been crossed to control which developmental pathway should be initiated [19,25,45]. Eventually, the initial photoperiodic signal is converted into a neuroendocrine signal, consistent with the presence or absence of JH peaks in the final-instar larvae. At the onset of the pupal stage, the JH signal is then relayed to target tissues such as the developing wings via the release (presence or absence) of ecdysone. If ecdysone is released within the first couple of days post-pupation, it potentially inhibits a default “*levana* gene expression profile” that leads to morph-specific activities such as the production of more red pigment ommatins in the wings. Chromatin regulation by epigenetic mechanisms such as histone modification, DNA methylation, and the expression of non-coding RNAs may subsequently fix the developmental program, leading to a determined phenotypic outcome [44]. A recent genome-wide study in the model lepidopteran *Manduca sexta* elucidated that complete metamorphosis is associated with profound transcriptional reprogramming mediated by epigenetic modifications such as DNA-methylation, involving approximately half of all the genes in this species [46].

In agreement with the above hypothesis, *A. levana* pre-pupae destined either for diapause or subitaneous development were shown to possess unique transcriptomic profiles consistent with season-specific adaptations [22]. These authors also identified a putative diapause duration clock gene expressed preferentially in diapause pupae, and several differentially expressed genes thought to play roles in the choice between seasonal phenotypes. More recent work has identified microRNAs (miRNAs) that contribute to the epigenetic control of polyphenism in *A. levana* [23]. Multiple miRNAs and their targets were shown to be differentially expressed in final-instar larvae and 1-day-old pupae. Because miRNAs predominantly have a negative effect on gene expression, they may inhibit genes required to generate the *A. levana* “standard morph” in response to environmental cues. Phylogenetic analysis suggests the *levana* morph is most likely the plesiomorphic (primitive) one, whereas the *prorsa* morph is apomorphic (derived) [32]. JH release in the final-instar larvae and/or early ecdysone release in the subitaneous pupae may, therefore, modify the expression of miRNAs that regulate the expression of genes necessary for both diapause and the formation of the *levana* phenotype (Figure 2). The examples below indicate how this may proceed at the epigenetic level.

For diapause maintenance, miR-289-5p (among others) is thought to silence the expression of genes related to metabolic processes during diapause in the flesh fly *Sarcophaga bullata* [47]. In *A. levana*, this miRNA was upregulated in larvae destined for diapause compared to those primed for subitaneous development, although there was no differential expression in the pupae [23]. If we assume that the properties of miR-289-5p as a diapause regulator are evolutionarily conserved, the following deduction would be reasonable. In *A. levana*, the initiation of gene expression responsible for metabolic arrest begins at the very end of the larval stage. A link to the JH signal in subitaneous final-instar larvae is also likely because its presence coincides with the significantly reduced expression of miR-289-5p at this stage. If true, JH may inhibit the expression of this candidate metabolic suppressor, which then ultimately leads to subitaneous development.

The regulatory mechanism that controls the duration of diapause and the determination of the adult phenotype is unclear, but preliminary and yet inconclusive experimental evidence points towards a diapause duration clock protein. In *Bombyx mori*, the ATPase TIME-EA4 measures time intervals and functions as a clock in diapausing eggs. It is important to determine the moment when diapause should be broken [48]. In *A. levana*, an ortholog of TIME-EA4—named the diapause bioclock protein (DBP)—was found to be strongly induced in *levana* pre-pupae destined for diapause [22]. Subsequent in silico target prediction provided evidence that DBP expression is regulated by miR-2856-3p [23]. However, this miRNA was also strongly upregulated in final-instar *levana* larvae when compared to subitaneous *prorsa* larvae, and the highest levels were reached in pupae representing both developmental pathways. This expression profile shows that DBP cannot be controlled by miR-2856-3p alone, but other epigenetic mechanisms such as histone acetylation or DNA methylation may well contribute to its regulation. Moreover, miRNAs can have hundreds of targets in insects [49] and further regulatory roles for miR-2856-3p, therefore, cannot be ruled out. Interestingly, when injected with an inhibitor of miR-2856-3p, final-instar subitaneous *prorsa* larvae displayed an intermediate *prorsa/levana* adult phenotype in 5% of tested specimens (Figure 3). Polyphenism-determining miRNAs have been reported in both hemimetabolous and holometabolous insects [50,51]. It is, therefore, conceivable that polyphenism in *A. levana* may also be controlled by a single miRNA master switch located directly upstream of the phenotype effector pathways, in which miR-2856-3p plays a key role but cannot be a sole determinant. Most likely, other epigenetic regulators are also involved.

In conclusion, these findings suggest a complex regulatory system controls diapause initiation, maintenance, and duration as well as the determination of the adult phenotype, with miRNAs such as miR-289-5p and miR-2856-3p as key components. However, the effects are not always consistent with a straightforward mechanism based on the inhibition of target effectors. Future research should, therefore, focus on the functional analysis of miRNAs and regulatory proteins, which will be facilitated by the availability of genomic and transcriptomic data as well as miRNA target predictions for both spring and summer generations [22,23,52]. Moreover, CRISPR/Cas9 genome editing is now feasible in lepidopteran species, making gene knockout studies in *A. levana* a promising approach [53]. Even though we still lack a complete picture of the regulatory network, the examples discussed above strongly suggest that both hormones and epigenetic mechanisms control the integration of environmental signals in *A. levana* to generate specific seasonal phenotypes.

## 4. Photoperiod-Specific Polyphenic Responses

### 4.1. Morphology

In addition to the visually striking wing polyphenism, the *A. levana* adult phenotype is affected by the photoperiod in other, more subtle ways, including biomechanical design. The *prorsa* imago is larger than the *levana* imago in absolute terms (larger, longer, less pointed wings, heavier thorax) and has a higher thorax muscle ratio, whereas the *levana* imago has a higher wing loading (i.e., fresh body mass divided by wing area) and a higher relative abdomen mass [35]. This is likely advantageous because *A. levana* tends to remain in the native habitat during the spring and invests in reproduction, whereas it disperses and expands its range in search of new habitats over the summer [35].

### 4.2. Immunity

The two *A. levana* morphs also show differences in larval immunocompetence, based on survival analysis following infection with the bacterial entomopathogen *Pseudomonas entomophila*, the antibacterial activity analysis in the hemolymph, and the expression of selected genes encoding antimicrobial peptides [54]. The *levana* larvae survived significantly longer than *prorsa* larvae, the antibacterial activity in the hemolymph was more potent, and genes encoding antimicrobial peptides were expressed at higher levels. Further analysis revealed that final-instar *levana* larvae also produce higher levels of phenoloxidase and lytic activity than *prorsa* larvae [55]. These results suggest a trade-off between immunity and other traits, such as reproduction. The seasonal adaptations of the distinct phenotypes of *A. levana* in terms of immunology can be explained plausibly by selective advantages. A more robust immune system in larvae committed to diapause [54,55] may benefit the pupae that are longer exposed to pathogens and parasites over winter.

### 4.3. Wing Pattern and Color

The most obvious differences between the seasonal phenotypes are the wing pattern and the color displayed by the spring and summer imagoes of *A. levana*, raising the question of why is it beneficial to be orange in spring and blackish in summer? The degree of melanization in insects has been attributed to environmental temperature adaptations. For example, the harlequin ladybird *Harmonia axyridis* (also known as the multicolored Asian ladybird) is a textbook example of polymorphism and polyphenism. It displays a number of distinct morphs differing in overall color and the number of the spots on the elytra. The black (melanic) forms (axyridis, conspicua, and spectabilis) are postulated to be advantageous in cold climates because dark surfaces absorb heat more quickly during exposure to sunlight. Such seasonal phenotypic plasticity allows individuals to produce the level of melanin necessary to maintain activity at the temperatures encountered when they emerge. In line with this hypothesis, it has been reported that cold temperature favors the darker morphs, and exposure of last instar larvae or pupae to an elevated temperature decreases the melanization of the *H. axyridis* beetles [56,57]. However, this widely accepted hypothesis does not apply to *A. levana* imagoes because their darker summer phenotype lives at elevated environmental temperatures when compared with the brown spring phenotype (Figure 1). A stronger melanization of the summer morph may result in a higher protection against pathogens and parasitoids, but the latter instead attack larvae and pupae of *A. levana*, which do not display differences in body color. Another possibility explaining seasonal polyphenism in *A. levana* butterflies could be distinct seasonal predation by insectivorous birds [58], resulting in adaptations related to camouflage, but there are no empirical data supporting this hypothesis.

## 5. Concluding Remarks

The development of *A. levana* is influenced by various environmental factors, including photoperiod, temperature, and seasonal changes in food quality [32,33,59]. However, the extent to which changes in body composition or life history traits are polyphenic adaptions to this largely predictable nutritional variation is unclear because most studies tested only the *prorsa* morph. The observed effects could also result directly from the diet quality, and this should be tested in future studies by co-varying light regimes and diets. In fact, a recent study by Esperk and Tammaru (2021) demonstrated that in terms of growth parameters, many between-generation differences have an adaptive nature and can, therefore, be considered part of the phenotypic response to seasonality. Similar study designs should be employed to scrutinize changes in body composition or life history traits reported in earlier studies.

The photoperiod takes overall precedence as the key climate signal that guides the decision between subitaneous and diapause development, but the outcome can be modified and fine-tuned by prevailing temperatures [24,25,26]. It will be interesting to see how *A. levana* responds to climate change, and this species could emerge as a useful bioindicator to track the increasing impact of rising temperatures on local ecosystems.

The photoperiodic signal is translated into physiological responses via hormonal regulation. If JH is released in the final-instar larvae, this is followed by early ecdysone release during the first few days of pupation, leading to subitaneous *prorsa* development. In contrast, the absence of JH delays ecdysone release such that the levels peak only after months of diapause, thus triggering *levana* development [19,40]. In addition to the striking differences in wing color and patterning, additional polyphenic distinctions include the details of biomechanical design and immunocompetence.

The perception of light signals (or their absence) and the subsequent storage and transmission of this information to the endocrine system and epigenome are not clearly understood. Despite considerable advances in the study of insect photoperiodism, the full pathway leading from photoreception to diapause and altered phenotypes is not known [43]. The mechanism of timekeeping and its integration should be prioritized when designing novel studies in this species. Epigenetic mechanisms such as the expression of miRNAs appear to play a key role in the transcriptional reprogramming that accompanies developmental commitment, with the *levana* morph seen as the default pathway and the *prorsa* morph as the alternative, but their position within the “environmental signal integration network” remains to be clarified [23].

## Figures and Tables

**Figure 1 insects-12-00325-f001:**
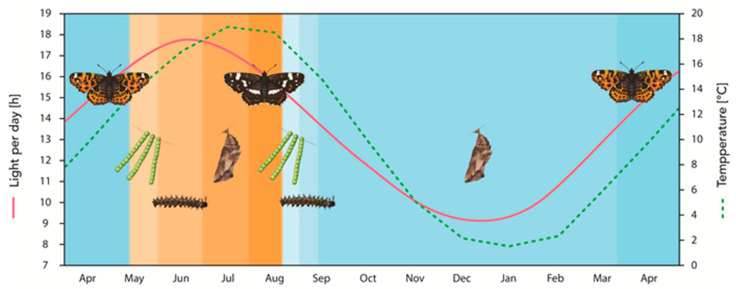
Annual life cycle and phenotype succession of *A. levana* depending on photoperiod and temperature. Values for day length and temperature correspond to the administrative district of Giessen, Hesse, Germany. Long day lengths (>15.5 h) and high temperatures (larval development from spring to midsummer) result in pupal subitaneous development and the expression of the adult *prorsa* form, emerging in summer. Conversely, short day lengths (<15.5 h) and low temperatures (larval development from late summer to early autumn) result in pupal diapause and the development of the adult *levana* form, emerging in spring. The brown color gradient represents *prorsa* development, blue color gradient represents *levana* development. The dotted line corresponds to the threshold day length of >15.5 h per day, below which *levana* development becomes increasingly likely. The developmental trajectory is not affected by environmental cues during embryonic development (green egg towers). For details, see text.

**Figure 2 insects-12-00325-f002:**
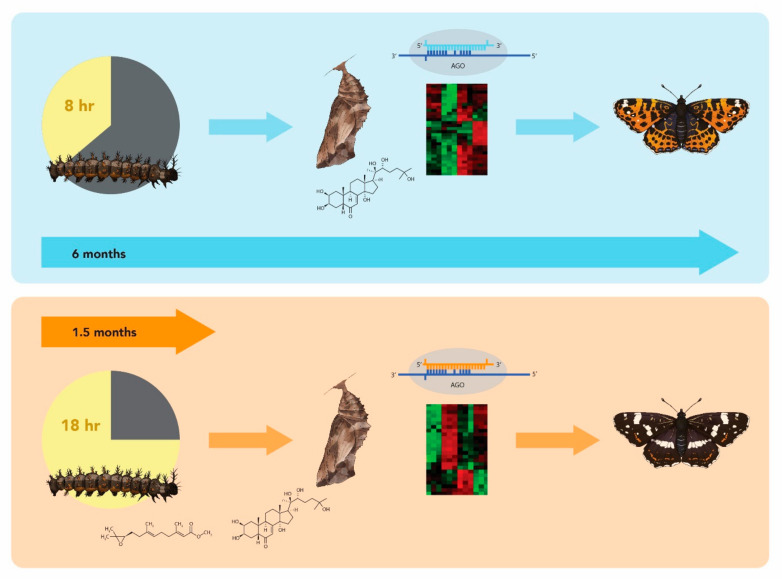
Model of development modes of *A. levana* according to photoperiod and temperature. At 20 °C *A. levana* develops directly (prorsa summer phenotype) if larval development occurs under long-day conditions (18 h photoperiod, lower half, life cycle lasting app. 1.5 months [thick orange arrow]) but switches to diapause development (levana spring phenotype, upper half, life cycle lasting app. 6 months [thick blue arrow]) if short-day conditions (8 h photoperiod) prevail. In the *prorsa* development path, juvenile hormone peaks just prior to pupation, followed by 20-hydroxyecdysone release within three days after the pupal molt. This initiates a gene regulation (e.g., by micro-RNAs, which are short non-coding RNAs of app. 22 nucleotides in length, that mediate gene silencing by guiding Argonaute (AGO) proteins to target sites in the 3′ untranslated region (UTR) of mRNAs [41]) and expression (symbolized by microarray) profile that results in direct metamorphosis onset and development of the adult *prorsa* phenotype. In the *levana* development path, the juvenile hormone signal is absent in the last instar and 20-hydroxyecdysone is not released until a cold period lasting at least 3 months has passed. Then gene expression results in the initiation of the imaginal molt and the adult levana form emerges.

**Figure 3 insects-12-00325-f003:**
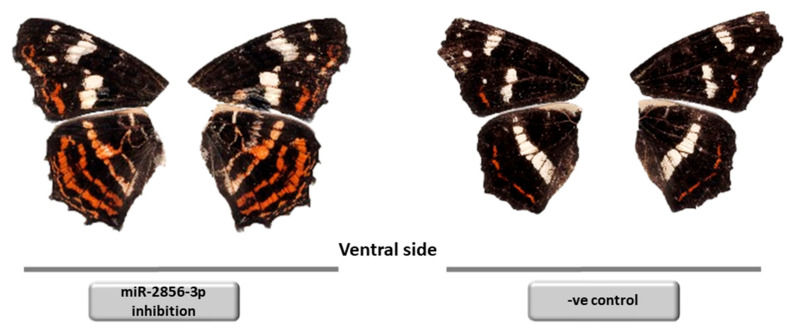
Intermediate and regular *prorsa* phenotypes of *A. levana*. The intermediate phenotype (**left**) was generated by injecting an inhibitor of *Bombyx mori* (bmo)-miR-2856-3p, whereas the normal phenotype (**right**) developed following the injection of PBS (control). The intermediate phenotype was observed in 5% of specimens [23].

## Data Availability

Not applicable.

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
