# Peer review of "The European Map Butterfly Araschnia levana as a Model to Study the Molecular Basis and Evolutionary Ecology of Seasonal Polyphenism"

_insects, 2021, doi:10.3390/insects12040325_

Round 1
Reviewer 1 Report
Without any doubt, this is a useful review on what is known about Arachnia levana, a species which has indeed served as a model object for various ecological and physiological studies. The potential of A. levana is definitely not exhausted so that this review will certainly inspire further studies.
I have the following suggestions to offer:
- I must say that I was slightly surprised by the fact that this review does not discuss the adaptive value of the polyphenism in wing pattern. From the point of view of evolutionary ecology, this is still the main question – the question ‘why?’ (in the evolutionary sense) is what defines evolutionary ecology, and you have ‘evolutionary ecology’ in the title. I understand that the emphasis of the paper is elsewhere but may you still provide a brief review of the ideas? Why is it good to be brown in spring and pied in summer?
- I would be happy to see more links to similar work on other butterflies/insects? Are there questions which are only/primarily/first studied in A. levana and not/less/later in other species? Which findings have been confirmed in other species? Are there things which may be unique to levana (different in other studied species)? May it be worth to compare the patterns in levana to those in other well studied polyphenic species, like the African Bicyclus butterflies?
- I would discuss all aspects of the polyphenic phenotype (i.e. also body composition and immunocompetence) in the introductory part of the paper, not at the end. Matter of taste, of course.
- I find that, on some occasions, you describe the work of other authors in too much detail (including exact numbers etc.). I would keep all this at a more general level. Matter of taste, once again.
- Just a few weeks ago, the following paper was published
Esperk, T. and Tammaru, T. 2021. Ontogenetic basis of among-generation differences in size-related traits in a polyphenic butterfly. – Frontiers in Ecology and Evolution, 9:612330. https://doi.org/10.3389/fevo.2021.612330
in which we compare larval growth schedules in the two seasonal generations of levana. I hope that you may find something relevant in that paper, at least it addresses the request you have in line 416: “this should be tested in future studies by co-varying light regimes and diets.”, this is exactly what we did.
Specific/minor
* line 46. “Phenotypic plasticity often facilitates adaptive changes by increasing phenotypic diversity in response to environmental challenges.” – sounds ambitious but is not very clear. “Adaptive value of phenotypic diversity” sounds like bet-hedging but this is not what it is about?
* line 68. “ventral wing” unusual, lepidopterists say “underside” or (less frequently) “ventral side”.
* line 17 “from scientists” – English OK?
* line 78 and elsewhere: were the works of Müller first to show responses to photoperiod in any insect? If yes, say this proudly. If not, tell which was first.
* line 85. Be more specific - when was it unclear? ... and do not start a sentence with But.
* Figure 1 text (and check elsewhere): do not say ‘increasing’ and ‘decreasing’ daylengths: there are insects which indeed respond to increasing (they understand that days are getting longer) and decreasing daylengths, this is not the case in levana as far as I know, or as you tell. Levana just responds to long and short days and does not sense the change?
* line 103. Now you tell the story of Müller twice: in the Introduction and here again. My suggestion is that you present less detail in the Introduction, this would help to reduce repetition.
* line 124. Perhaps you mean MORE THAN 15.5 h, it cannot be invariable for such a long period. And of course this crucially depends on latitude which you have not specified.
* line 148-149. I would delete this sentence.
* line 157. 22]. This indicates THAT there is a point...???
* lines 167-175. This is not very clear, please clarify!
* The “Food quality” chapter. Please make a clear connection to the seasonal polyphenism topic from the beginning of this chapter. Now the reader wonders why are you telling all this here… and the connection is now made only at the end of the chapter.
* line 237 less THAN their?
* line 286. determination changes?
* Figure 3. Why do you show prorsa and prorima but not levana? I would show all three.
* line 395. Evolutionary ecologists usually do not say “is advantageous because” like it was for sure, people usually add an element of doubt to such adaptive stories. And perhaps native habitat, not local?
* line 412. “The development of A. levana is influenced by various environmental factors, including photoperiod, temperature and seasonal changes in food quality [29,30,53]. However, the extent to which changes in body composition or life history traits are polyphenic adaptions to this largely predictable nutritional variation is unclear because most studies testedonly the prorsa morph.” Here the logic is not very clear, how can one know anything about the difference if only one morph was studied? Please clarify!
* 421. What exactly is levana expected to indicate in the context of climate change? Sounds little hollow.
Good luck!
Reviewer 2 Report
This review paper summarizes the findings on seasonal polyphenism seen in Araschnia levana, which is famous for very different wing color patterns between two seasonal morphs. Generally, this manuscript is well written. Uniquely, this review often mentioned German papers, which are not readily accessible for researchers outside Germany. I can only comment on the following points.
Major points
1) This paper focuses exclusively on levana. But there are studies that use other lepidopteran species such as Bicyclus anynana for similar objectives. Such studies on lepidopteran seasonal polyphenism should be briefly mentioned in Introduction or in a new section.
2) It is advisable to discuss biological function of this polyphenism in Introduction, even if such function has not been established well.
3) Figure 1 is nice, but it is unclear which locality this light and temperature diagram come from (likely somewhere in Germany). Authors should indicate exact locality from which light and temperature data were obtained for this figure.
4) Figure 1: Brown and blue color gradations need an explanation.
5) Figure 1: This figure would be more informative if threshold daylength is included in this figure.
6) Figure 2: What is AGO?
7) Figure 2: I do not understand an arrow with a label of “1.5 months” and an arrow with a label of “6 months”. Do they indicate the larval periods or a single life cycle?
8) The last part of page 9 is confusing. The intermediate phenotype was obtained only in 5% of treated individuals. This means that the full change to the levana phenotype was not attained in any individual. To me, this result indicate that this miRNA is not a sole determinant. Please clarify this point.
9) Figure 3 is too large. Please make it much smaller.
10) Only right or left side is enough (and indeed better for comparison) for Figure 3.
11) In the Figure 3 legend, miRNA-2856-3p was described as bmo-miR-2856-3p. What is this “bmo”?
Minor points
L22: “talk about” is for oral expression. Simply use “discuss”.
Keywords may include phenotypic plasticity.
L394: What is “wing loading”?
L431 “is” should be “are”.
Reference 9: Is full citation possible including publisher?
Reference 52: “Front. Physiol.” should be italic.
